# Response Characteristics of Electric Potential and Its Relationship with Dynamic Disaster during Mining Activities: A Case Study in Xuehu Coal Mine, China

**DOI:** 10.3390/ijerph19158949

**Published:** 2022-07-23

**Authors:** Yue Niu, Zhonghui Li, Enyuan Wang, Tiancheng Shan, Heng Wang, Shilong Xu, Wenyang Sun, Guanteng Wang, Xingzhuo Xue, Junqi Liu

**Affiliations:** 1State Key Laboratory for GeoMechanics and Deep Underground Engineering, China University of Mining and Technology, Xuzhou 221116, China; y.niu16@cumt.edu.cn; 2Key Laboratory of Gas and Fire Control for Coal Mines, Ministry of Education, China University of Mining and Technology, Xuzhou 221116, China; shantiancheng@cumt.edu.cn (T.S.); ts21120051a31@cumt.edu.cn (H.W.); 02203709@cumt.edu.cn (S.X.); 16214920@cumt.edu.cn (W.S.); 02203710@cumt.edu.cn (G.W.); 02205203@cumt.edu.cn (X.X.); 02205001@cumt.edu.cn (J.L.); 3School of Mechanics and Civil Engineering, China University of Mining and Technology, Xuzhou 221116, China; 4School of Safety Engineering, China University of Mining and Technology, Xuzhou 221116, China; 5National Engineering Research Center for Coal Gas Control, China University of Mining and Technology, Xuzhou 221116, China

**Keywords:** deep coal seam, mining activities, electric potential response, dynamic disaster, safety monitor

## Abstract

Across the world, coal resource is widely utilized in industrial production. During coal mining activities, dynamic disasters may be induced, such as coal and gas outbursts, or rock burst, resulting in serious accidents or disasters. Previous studies have shown that electric potential (EP) signals can be produced during the deformation and fracture process of coal and rock mass under load. The abnormal response characteristics of EP can reveal the damage evolution and failure feather of coal mass. In this paper, the response characteristics of EP signals are analyzed with high gas testing during mining activities within deep coal seams, and the relationship between the EP response and outburst disaster hazard is studied. The results show that: (1) Under the comprehensive action of mining stress and gas effect, the coal mass was damaged and fractured, which can produce abundant EP signals, while the temporal EP response characteristics can reflect the loading state and damage evolution process inside the coal seam. (2) When coal cannon and a sudden increase of gas concentration occurred in the coal mass, the EP signal was at a high level and fluctuated violently. This can be regarded as precursory information for an outburst risk, which was verified by monitoring the results of mining stress and electromagnetic radiation (EMR). (3) Based on the unilateral inversion imaging method, EP spatial distribution law was studied and abnormal zones with high-value were identified. The zone is close to, or coincident with, the high value interval of EMR intensity and count indexes, which revealed the distribution characteristics of coal damage localization. Hence, EP monitoring results can forecast precursor information of outburst hazards temporally, and identify local zones with outburst hazard spatially. This study provides a new idea and application basis for using the EP method to monitor and prevent coal and rock dynamic disaster hazards in the field.

## 1. Introduction

Coal is a major energy resource, widely utilized in industrial production and daily life, such as power generation, supply heating, steel smelting, and coal chemical synthesis et al. [1]. Annually, more than seven billion tons of coal are mined from the earth and consumed by human beings all over the world. Among them, China accounts for more than half, with the electricity generated coming from thermal power (energy obtained by burning coal). With the increasing depletion of shallow coal resources, coal mining has entered the deep mining stage [2,3]. Considering the environment of high in-situ stress, high gas and complex geological structures deep underground, the threat of dynamic disaster is more serious in the production of mining activities, such as coal and gas outburst, or rock burst et al. [4,5,6,7]. For example, on 10 June 2020, a serious coal and gas outburst accident occurred in Liaoyuan, China, which resulted in seven deaths and two injuries, with a direct economic loss of 16.66 million yuan. Hence, it is significant to carry out real-time monitoring and fine forecasting of coal and rock dynamic disasters during coal mining activities [8,9,10].

The main cause of dynamic disaster is the destruction of coal and rock mass. Essentially, this is the result of deformation and fracture of coal and rock mass, which leads to continuous damage evolution under the comprehensive action of mining stress and gas effect [11,12,13]. Previous studies show the deformation and fracture process of coal mass can generate electric potential (EP) signals. The EP response is closely related to the stress state, and the evolution process of damage and failure feather of coal and rock mass [14,15,16,17]. Among them, Eccles et al. observed EP precursor signals before sandstone’s failure in the experiment [18]. Wang et al. studied the different EP effect between coal and rock specimens under different loading styles, and revealed the microscopic mechanism of EP generation [19]. Niu et al. developed the constitutive model between damage evolution and EP response of gas-bearing coal, and calculated the effective stress of coal mass based on EP data [20]. Li et al. found that under the coupling action of stress and gas, EP signals have a precursory characteristic of abnormal response, before the failure of the coal specimen [21]. The above research results show it is promising to monitor the damage evolution process, and forecast the failure and catastrophe characteristics of coal mass with the EP method, under the joint influence of stress and gas. Considering the monitoring advantages of accurate response and strong anti-interference for signal measurement, the EP method is expected to be applied to the monitoring and forecasting of dynamic disasters during mining activities [22].

However, previous researchers have mainly focused on theoretical research and the experimental analysis of EP’s effect on coal and rock mass. Little research has been carried out in the area of coal mines, especially with regards to the driving activities of high-gas coal seams in deep coal mines. In essence, the driving activity within coal seams induces change within the original stress equilibrium state of coal and rock mass, resulting in its deformation and damage under the comprehensive action of mining stress and gas effect, which leads to the hazard of dynamic disaster occurrence [23]. In theory, the hazard of dynamic disaster should be closely related to the mining stress state [24]. Simultaneously, according to mining activity experience, there are some precursor characteristics which occur before dynamic disasters, such as the coal cannon phenomenon and rapid increase in gas concentration (for example, such as a coal and gas outburst) [25]. In addition, conventional monitoring indicators can also be used for forecasting of dynamic disasters, such as electromagnetic radiation (EMR), microseismic and drilling cuttings [8,26,27].

This paper will therefore compare the advantages and disadvantages of the above EP monitoring, selecting a high-gas coal seam within a deep coal mine, in order to monitor the EP response characteristics during driving activities (one kind of major mining activity). In addition, the time correspondence between EP response and the precursory characteristics of an outburst hazard were analyzed. Further, EP spatial distribution law was studied, and the corresponding relationship in the space between abnormal EP zones and outburst hazard zones, were analyzed. The research results are expected to provide new ideas and lay a foundation for further using EP to monitor and forecast dynamic disasters during mining activities.

## 2. EP Monitoring Scheme of Coal Mass around Driving Face in Coal Seam

### 2.1. Overview of the Driving Face

As shown in Figure 1, 25,020 Air Roadway was selected for testing in the 25 Mining Area of II_2_ Coal Seam, Xuehu Coal Mine, Henan Province, China. During the testing process, driving activities were in progress. In the coal seam, the maximum gas pressure and maximum gas content were 0.415 MPa and 8.6483 m^3^/t, respectively; the initial velocity of gas emission from the coal seam was 12.957~14.000 Pa; and the comprehensive indexes D and K were 10.6745~11.1687 and 31.9216~63.6364, respectively. All these are indicative of belonging to high-gas coal seam.

Meanwhile, there were some faults developing. The mining stress was high, consequently, abnormal stress zones were easily formed in local areas. This may lead to dynamic failure and even dynamic disasters, such as roof cracking, fracture development and gas enrichment. The coal seam consisted of a coal and gas outburst hazard; therefore, attention should be given to the monitoring and forecasting of outburst hazard during driving and mining activities in the coal seam [28]. 

### 2.2. EP Test Schemes

Developed by the EMR research group in China University of Mining and Technology [29], an EP monitoring device for mining was utilized for the test scheme, which used thin, rectangular copper electrodes. Firstly, a borehole was drilled according to the measuring position and the electrode was placed at the bottom of the hole. The electrode was connected to the device host through a wireway, so that the EP signal inside the coal seam could be measured by the electrode.

The EP test scheme included two parts. The first was to arrange EP measuring points ahead of the driving face, in order to measure changing EP signals inside the coal mass, along with the distance from each measuring point to the driving face. Consequently, the relationship between EP signals and mining stress (MS) can be analyzed. The second part was to arrange measuring points behind the driving face, in order to measure EP signal changes to the driving time, so that the response characteristics of EP signals as precursors to outburst hazards could be accurately studied.


**(1).** 
**Test scheme of measuring points ahead of driving face**



As shown in Figure 2a, test borehole A was drilled inside the coal mass ahead of the driving face. The borehole was inclined by 13°, relative to the coal wall of the driving roadway, while the borehole orifice had a distance of 2 m from the driving face. The borehole bottom had a vertical distance of 5 m from the roadway wall, and a horizontal distance of 2 m from the driving face. Measuring points of EP and mining stress were arranged near the borehole bottom, with a spacing of 1 m. Consequently, during driving activities the changes in EP signals and mining stress could be measured inside the coal mass ahead of the driving face.


**(2).** 
**Test scheme of measuring points behind driving face**



As shown in Figure 2b, test borehole B was drilled at a position 2 m behind the driving face, positioned vertically from roadway wall. The measuring points of EP and mining stress were separately arranged at borehole depths of 4 and 5 m, to monitor the changes of EP signals and mining stress inside the coal mass, behind the driving face during driving activities.

## 3. Sequential Monitoring Results of EP Responses of Coal during Driving Process

### 3.1. EP Response Characteristics of Coal ahead of Driving Face

When arranging EP measuring points in the coal mass ahead of the driving face, the distance from the EP measuring points to the driving face continually shortens with the advance of the driving face. The distance is defined as “driving distance” between EP measuring points and the driving face. The EP measuring results are shown in Figure 3.

Figure 3a shows the changes in EP signals along with driving time, which show a trend of stable increase, fluctuations at high values, and a rapid decrease with the advance of the driving face. In the early measuring stage, EP signals gently increased. Around 24 December, however, EP signals reached their peak value and then dropped abruptly, showing remarkable fluctuations within this stage. With further advancing of the driving face, the EP measuring point gradually approached the driving face, and EP signals reduced significantly. Moreover, the daily EP mean value was calculated in Figure 3a. The driving distance of the corresponding day was also recorded, in order to obtain changes in the EP mean value with the driving distance. In addition, the EP mean value was analyzed, combined with mining stress, as displayed in Figure 3b. As the driving face was advanced further, the EP mean value and mining stress both changed in a trend which first ascended, reaching high values, and then suddenly descended. When the mining stress was in an interval of high values and large changes, the EP mean value was also in a high value interval. Compared with the sequential EP signals, the changes in the EP mean value shows better regularity.

According to analysis of the above results, original stable structures in coal mass are disturbed and damaged during the driving process, so that mining stress is redistributed. The non-uniform changes in stress induce non-uniform damage to the coal, thus emitting EP signals [30]. Meanwhile, damage to the coal causes change to the coal fracture field, inducing gas diffusion and migration, which further accelerates and promotes damage and fracture of the coal, and aggravates EP responses and fluctuations [31]. The coal is more prone to suffer from plastic deformation and fracture under higher mining stress. The plastic deformation of coal can also induce intense fluctuations of mining stress in a high value interval, which in turn aggravates damage to the coal. Consequently, the coal suffers intense destruction when the mining stress changes substantially, or stays at a high level persistently, as shown in Figure 3 [32]. Hence, the EP signals are high and fluctuate violently, showing significantly abnormal response characteristics. 

In accordance with the classical distribution law of mining stress, the stress state inside coal mass ahead of the driving face gradually experiences a transition from an original stress zone, to a stress concentration zone and then to a pressure relief zone, with the advance of the driving face [33,34]. Meanwhile, the sequential EP signals stabilize at first, then increase, before finally decreasing, which indicates high consistency between changes in EP signals and mining stress. Therefore, sequential responses of EP signals can reflect the coal’s stress state and damage evolution process.

### 3.2. EP Response Characteristics of Coal behind the Driving Face


**(1).** 
**Sequential EP Monitoring Results**



When arranging EP measuring points in the coal behind the driving face, the distance from the measuring points to the driving face increases constantly with the advance of driving face. EP measuring results are shown in Figure 4. 

In the early stage of the driving process, EP signals fluctuated violently with high amplitudes, before fluctuating relatively regularly. Around 5 and 10 December, abnormal EP response interval occurred with high EP amplitudes and multiple high-impulse responses. On 7 and 9 December, high value intervals of EP responses appeared and EP peaks were high, with remarkable fluctuations; whereas abnormal response characteristics were relatively weaker than those of 5 December and 10 December.

According to analysis of the above results: ① In the early driving stage, when EP measuring points are close to the driving face, the driving face can be regarded as a stress relief space for the coal ahead, under the influence of driving activities. As a result, the pressure relief effect and stress concentration effect appeared in coal behind the driving face, so the coal was substantially damaged, and the mining stress fluctuated accordingly. Therefore, the EP signals changed violently, and had high values. ② According to research records, around 5 and 10 December, gas concentration rose suddenly at EP measuring points. The rapid gas emission aggravated the damaged and fractured coal, indicating an outburst hazard. On 9 December, high-energy coal cannon events were recorded in the driving roadway, while huge elastic energy stored in the coal was released, which induced dynamic failure in local areas, showing certain outburst hazards. As displayed in Figure 4, abnormal response characteristics of EP signals occurred.

**(2).** 
**Comparison of EP Mean Value and Mining Stress**


The daily EP mean value was calculated to analyze the relationship between EP mean value and mining stress value, with the driving distance (shown in Figure 5). When the distance from measuring points behind the driving face, to the driving face, is 12~19 m, the mining stress changes violently and the EP mean value is at a high level. If the distance is shorter than 12 m, the EP signal fluctuates violently, while the EP mean value in the corresponding days is still low.

According to analysis of the above results, when the mining stress is high, but less than the failure strength of coal mass, theoretically, the coal is still subjected to plastic deformation rather than deformation and fracture. Compared with when mining stress is high, the remarkable changes in mining stress are more likely to induce damage to the coal, so the EP mean value is at a high level. As illustrated in Figure 5, although the mining stress is high when the measuring points are less than 10 m from the driving face, it does not change violently, consequently the EP mean value is also low.

**(3).** 
**Comparison Results of EP and EMR Measurements**


The EMR monitoring technology is a routine method for monitoring and forecasting of coal and gas outburst hazards, included in the official guidelines for the safe production of coal mines in China. When EP measuring was carried out, the EMR indexes (namely, intensity and count) of the coal mass around the driving face were also monitored. In this scheme, the EMR signal was measured with the equipment of KBD 5 [35]. The measurement method was non-contacting. When the EMR measuring point was selected and confirmed, as shown in Figure 2b, the antenna was closely aimed at the coal mass (EMR point) on the coal wall, without touching it. The EMR signal produced by coal mass on the EMR point could then be measured and transferred to the EMR host with a wireway. While the EMR monitoring process is similar to EP, the difference is there is no touching.

The EMR changes with the driving distance are displayed in Figure 6. The EMR count index reflects the occurrence frequency of micro-fracturing events inside coal mass, while the EMR intensity reflects the energy release value from coal, due to these micro-fracturing events.

According to analysis of the above results, the EP mean value changes with the driving distance basically in a trend (decrease-increase-decrease-increase-decrease), which is the same as the EMR intensity and count. ① When the distance was less than 10 m from the measuring points to the driving face, the EP mean value and the EMR indexes were both in a high value interval. This indicates that the coal was seriously deformed and damaged in the interval under driving influences. ② If the distance was 18~23 m, the EP mean value was in the high value interval and the EMR count was high, while the EMR intensity changed significantly from high to low value. This was because the micro-fracturing of coal is of low mechanical intensity; the micro-fracturing signals were rapidly attenuated in the coal medium. EP monitoring was a contact-monitoring method between electrodes and coal mass, while EMR monitoring was non-contact, so the EP index obtained in the contact monitoring route showed more significant response characteristics, compared with EMR indexes. ③ When the measuring points were 28.8 and 40.2 m from the driving face, outliers occurred to both the EP and EMR indexes. According to previous analysis, this was caused by rapidly increased gas concentration.

### 3.3. Summarization of Regularities in Sequential EP Monitoring Results

The above analysis indicates that EP signals exhibit abnormal responses correspondingly when the coal was subjected to dramatic damage in local areas, during the driving process. Especially when the coal was at risk of dynamic failure, such EP response characteristics became more significant. When the EP mean value was high, or the EP signal showed abnormal and abruptly changed responses, it generally meant violent damage to coal mass in local areas. The sequential response of EP signals was able to reflect the stress state and damage evolution inside the coal mass.

For the coal seam with a high gas content, damage occurred to coal mass under the comprehensive action of mining stress and gas effect [36]. EP responses resulted from deformation and fracture of coal under the effect of mining stress and gas, rather than being only related to the loading state of mining stress. This explains why the mining stress remained relatively stable when abnormal EP responses occurred. When the mining stress was relatively stable, the action of gas on coal became a key factor that intensified damage and fracture of the coal mass, inducing abnormal EP responses.

In comparison, the action of gas was likely to be ignored in the monitoring of mining stress, so the outburst hazard could not be completely revealed in the coal seam [37]. Although EMR monitoring can comprehensively reflect the frequency and intensity of micro-fracturing events during damage to coal, EP monitoring as a contact monitoring method can provide more sensitive response signals. EP signals can more comprehensively show the mining stress state, revealing the damage evolution process, and therefore monitor the outburst hazard.

## 4. Spatial Distribution of EP signals of Coal during Driving Process

### 4.1. EP Inversion Imaging Method

The previous statistical results show that the area where outburst disasters occur in coal seams was generally concentrated locally, and the area of outburst hazard only accounted for a low proportion of the whole coal seam. Therefore, the area of outburst risk needs to be precisely identified in the driving face. Under the coupling of stress influence and gas effect, the coal and rock mass deforms and fractures, causing free charges and charge separation [19]. Coal mass damage and fracture can be regarded as the point source for abnormal charges, and produces EP signals on the boundary of EP measuring lines. The cloud map for distribution of charge sources in the coal can be obtained by computed tomography through inversion, using geophysical methods based on EP signals on the boundary of measuring lines [38]. This indirectly reveals the spatial distribution of EP signals. High value areas correspond to the locations of abnormal charges in coal in the distribution results of EP inversion values (absolute values in the interval of 0~1) in the cloud map, where coal is severely damaged and can be deemed as an outburst risk.

The unilateral EP inversion imaging method was utilized. By testing EP data at multiple points, the continuous spatial distribution of EP signals in a certain area of coal was obtained to identify the area of coal and gas outburst risk. The results were also verified by combining the EMR monitoring results of coal in the area. As shown in Figure 7, there were 11 EP measuring points arranged on the coal wall roadway behind the driving face, with a spacing of 5 m. The inversion area was a rectangle, which was 50 m behind the driving face and 10 m inwards from the coal wall to the solid coal direction. The EP inversion imaging method of coal has been deduced and demonstrated in details in previous research [39]. In brief, EP values were tested to calculate the EP difference and spacing between adjacent measuring points, and to further solve component distribution of the electrical field intensity, at different measuring points, as shown in Figure 7. By constructing the scanning function, the EP inversion value can be calculated at any point in the inversion area. Then, the continuous cloud map of EP inversion values can be drawn through the Kriging interpolation method of the whole area.

### 4.2. Spatial Distribution of EP Inversion Results

The cloud map of EP signals in the inversion area of coal behind the driving face was obtained through measuring and imaging inversion, as displayed in Figure 8a. The dashed white lines indicated the zones with high EP inversion value. The arrow indicated the starting position of the EP inversion area (where the abscissa value was zero), meanwhile, the arrow direction indicated the increasing direction of the ordinate value. The gray area represented the driving roadway, and the cloud map of EP inversion showed the solid coal area outside the roadway. Because EMR signals were obtained by non-contact monitoring that failed to reach the deep part of the coal, the EMR indexes were only monitored on the coal wall roadway in Figure 8a, to verify the EP inversion results, as shown in Figure 8b. The red, black, and pink rectangles indicated the high zones of EMR count, EMR intensity and EP inversion values. The boxes enclosed by black, red, and pink lines, separately denote the high value intervals of EMR intensity, EMR count and high value intervals in the cloud map of EP inversion.

In the cloud map of EP inversion, there were three clear high value areas: the highest value area, secondary high value area, and relative high value area, which were 18~26 m, 31~44 m, and 7~11 m from the driving face, respectively. In the cloud map of EP inversion, the highest value area (0.272~0.308), secondary high value area (0.308~0.600), and relative high value area (0.272~0.488), which were 18~26 m, 31~44 m, and 7~11 m from the driving face, respectively. Correspondingly, in the first local high values (from left to right in Figure 8b), the EMR intensity range was from 54 to 58 mV and EMR count was from 64 to 70. Moreover, in the second local high values, the EMR intensity range was from 91 to 106 mV and EMR count was from 113 to 125. Further, in the third local high values, the EMR intensity range was from 105 to 120 mV and EMR count was from 97 to 104. 

The three areas differed significantly in the EP inversion value, indicating that the coal seam was characterized by significant localized damage. As shown in Figure 8b, high value areas of EMR intensity and count were slightly overlapped, but were close to each other. The high value areas in the cloud map of EP inversion were basically overlapped with the whole high value areas of EMR indexes. A fault occurred at the location 23~26 m behind the driving face. The presence of the fault impaired the intactness and continuity of the coal seam. The tectonic stress was superimposed with mining stress at the fault, causing more violent damage to the coal. This was observed in the area with local high values of EMR intensity and count, while peak EP inversion value (up to 0.6) was found in the area of cloud map with dense isolines. Serious spalling of the coal wall (namely, “wall caving”) was observed in the area 30~35 m behind the driving face, indicating occurrence of stress concentration in the coal, and violent local damage. Local high values of EMR intensity and count were found in the area around the spalling, where the secondary high value area appeared in the cloud map of EP inversion. 

The analysis results show that the high value intervals of EMR intensity and count are close to, or overlapped, behind the driving face, while the high value areas in the cloud map of EP inversion were basically overlapped with the high value areas of EMR indexes. The result indicated that areas of dense and high-intensity micro-fracturing events could be identified in the coal seam based on EP inversion results. The localized damage characteristic of coal mass was reflected in the abnormal areas of EP inversion values. The abnormal areas (high value areas) in the cloud map generally mean that local areas of coal were subjected to violent damage and were key areas for monitoring outburst hazard. If the cloud map of EP inversion was distributed uniformly, without significant high value areas, it indicated relatively uniform damage and deformation of the coal seam under the influences of driving, and the absence of areas of obvious outburst hazard.

## 5. Discussion

### 5.1. Relationship between EP Response Characteristics and Outburst Hazard

When driving in a deep coal seam with a high gas content, there are certain hazards that induce coal and gas outburst disasters [37]. Coal cannon and an abrupt increase in gas concentration can be regarded as precursors for outburst hazard. Among them, coal cannon refers to the accumulation of an abundant strain energy inside coal mass under the stress concentration effect, and subsequently a sudden release induced by disturbance after reaching a high degree. Coal cannon generally occurs instantaneously, causing serious local damage to coal, which threatens the safety of personnel and equipment in the driving space, and even further induces coal and gas outburst accidents. When coal cannon happens, EP signals and the EMR intensity and count all have abnormally high values. When the coal structure stabilizes, gas adsorption and desorption reach dynamic equilibrium. If the local internal part of the coal is damaged violently, and dynamic failure is induced, the internal structure forms a free space, which opens gas diffusion and migration paths inside the coal mass, enabling an inrush of abundant gas from the coal to the driving roadway. Therefore, the rapid increase in gas concentration indicates that the internal part of the coal is undergoing violent damage, and the rapid gas emission tears existing fractures and channels in the coal, further aggravating damage to it. Meanwhile, the EP signals and EMR indexes all grow rapidly. As gas concentration increases substantially to the critical value, the coal seam is at risk of outburst accidents; therefore, prevention measures should be taken immediately, and their effects should be checked simultaneously [23].

According to analysis in Section 3, coal is damaged under the joint action of driving activities and gas effects, thus producing EP signals. Therefore, EP response characteristics can reflect the stress state and damage evolution process of coal mass [15]. When coal cannon happens, or gas concentration rises abruptly, the EP value rapidly increases to a high level and fluctuates remarkably. These can be regarded as precursors and forecasting information for coal and gas outburst risks, and have been verified by monitoring the results of mining stress and EMR indexes. Section 4 reveals that areas in the driving face, that are truly at risk of outburst, is generally concentrated locally due to the influences of mining stress distribution in the coal seam, geological structures, and non-uniform gas emission. The spatial distribution of EP signals also shows significant local concentration characteristics. Consequently, the EP inversion imaging results can reveal the spatial distribution of EP signals in coal around the driving face, and reflects localized characteristics of damage to coal mass [39]. By identifying abnormal areas in the cloud map of EP inversion, the areas with risk of coal and gas outburst can be identified, for which targeted measuring can be taken with regards to prevention and control.

### 5.2. Application Prospect of EP Monitoring for Forecasting Coal and Rock Dynamic Disaster Hazard

EP monitoring has its own unique advantages, compared with traditional monitoring methods for coal and gas outburst. On the one hand, EP signals are highly responsive. They are not only sensitive to large fracturing events in coal, but also respond to changes in the deformation, that is, the strain field of coal [40]. Therefore, EP signals can profoundly reveal the damage evolution process of gas-bearing coal. On the other hand, EP monitoring performs better in resisting interference. EP signals are measured by direct contact between electrodes and coal [14]. Coal mass is a poor conductor. Compared with EMR signals, EP signals in deep coal are slightly affected by the operation of electromechanical equipment, so EP signals have a stronger interference resistance to background electromagnetic signals in the working environment. This research provides a new monitoring method for accurately identifying coal and gas outburst risks, and lays out a research basis for its application.

## 6. Conclusions

In this paper, the temporal characteristics and spatial distribution of EP signal responses were measured and analyzed, which were generated during the driving activities in a deep, high-gas coal seam. Further, the relationship was studied between EP response characteristics and the outburst risk. The following conclusions were obtained:(1).Under the comprehensive action of mining stress and gas effect, the coal mass could generate abundant EP signals during the driving process. The temporal response of the EP signals could reflect the loading state and damage evolution process of coal mass. When the mining stress was at a high level, or when the mining stress changed drastically, the coal mass was severely damaged and fractured, while EP intensity was at a high value level. When the mining stress was relatively stable, the gas effect on coal mass was key to aggravating the damage and breakage of the coal mass, which induces abnormal response of EP signals.(2).When the coal cannon phenomenon occurred, abundant elastic energy was accumulated inside the coal mass, which subsequently was suddenly released under the driving disturbance. Consequently, the EP signals present an abnormally high value. When the gas concentration increased rapidly, the interior coal mass suffered local severe damage, and the EP signals increased rapidly. Accordingly, the EP signals increased rapidly to a high value and fluctuated violently, which was regarded as precursory forecasting information of coal and gas outburst disaster risk. This was also verified by monitoring results of mining stress and EMR index.(3).Based on the unilateral inversion imaging method, the EP spatial distribution law in a certain region can be obtained with the help of multi-point EP test results. The distribution of the EP inversion cloud map has obvious and significant local concentration characteristics. The abnormal zones are close to, or coincident with, the high value interval of EMR intensity and count, identifying damage localization areas in the coal seam during driving activities. Through the identification of abnormal zones in the EP cloud map, it can identify the risk zones of coal and gas outburst. These results lay the foundation for taking targeted measures to prevent and control dynamic disasters.

## Figures and Tables

**Figure 1 ijerph-19-08949-f001:**
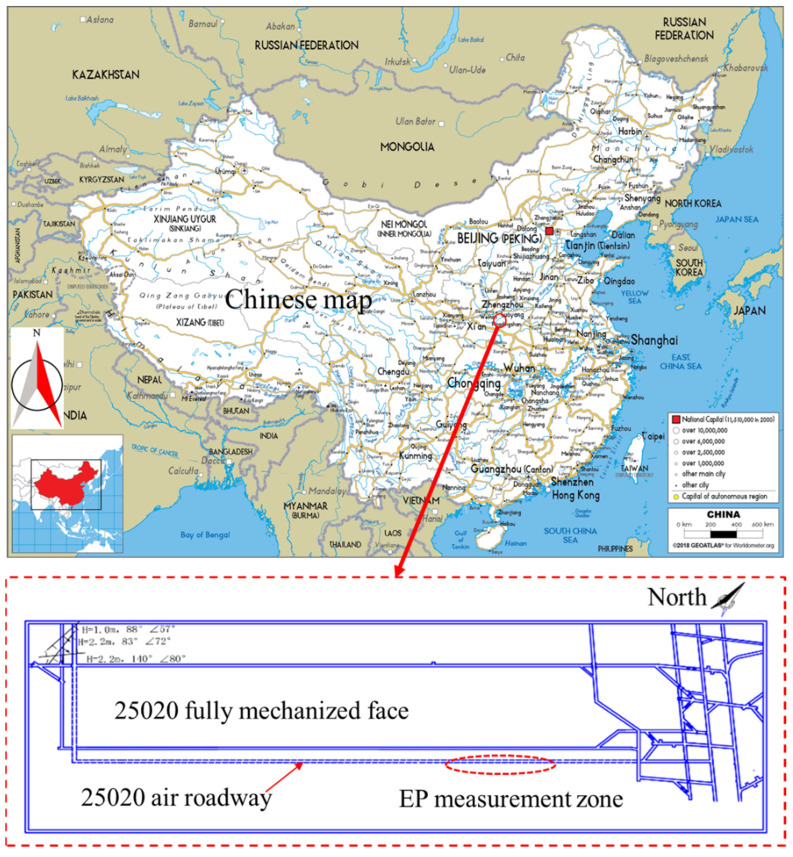
Schematic diagram of EP measurement zone in 25,020 Air Roadway, Xuehu Coal Mine.

**Figure 2 ijerph-19-08949-f002:**
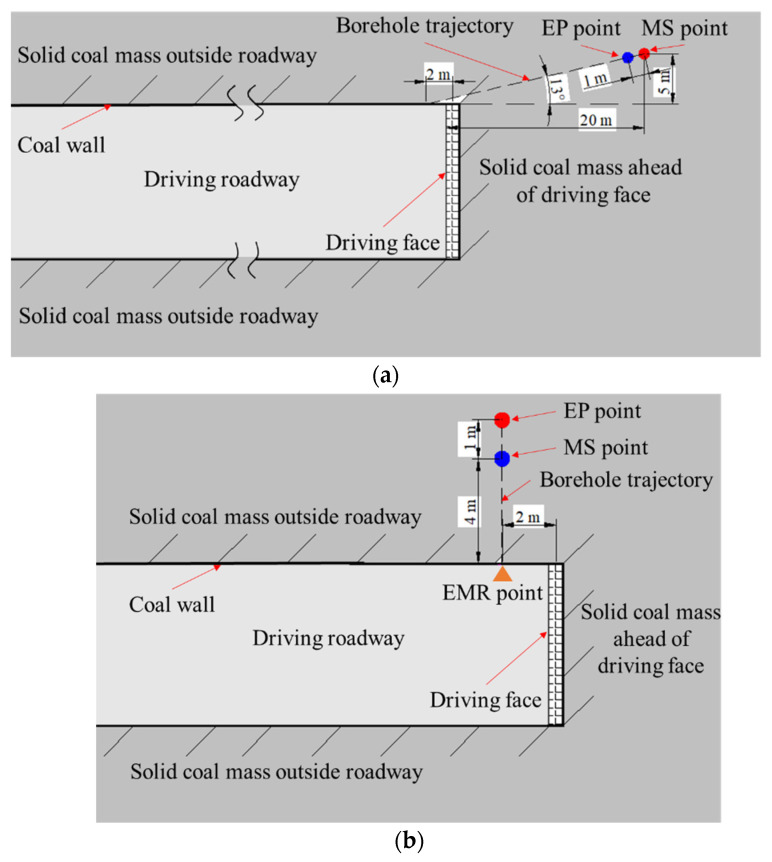
Planar diagram of EP measuring points around the driving face. (**a**) Ahead of the driving face; (**b**) Behind the driving face.

**Figure 3 ijerph-19-08949-f003:**
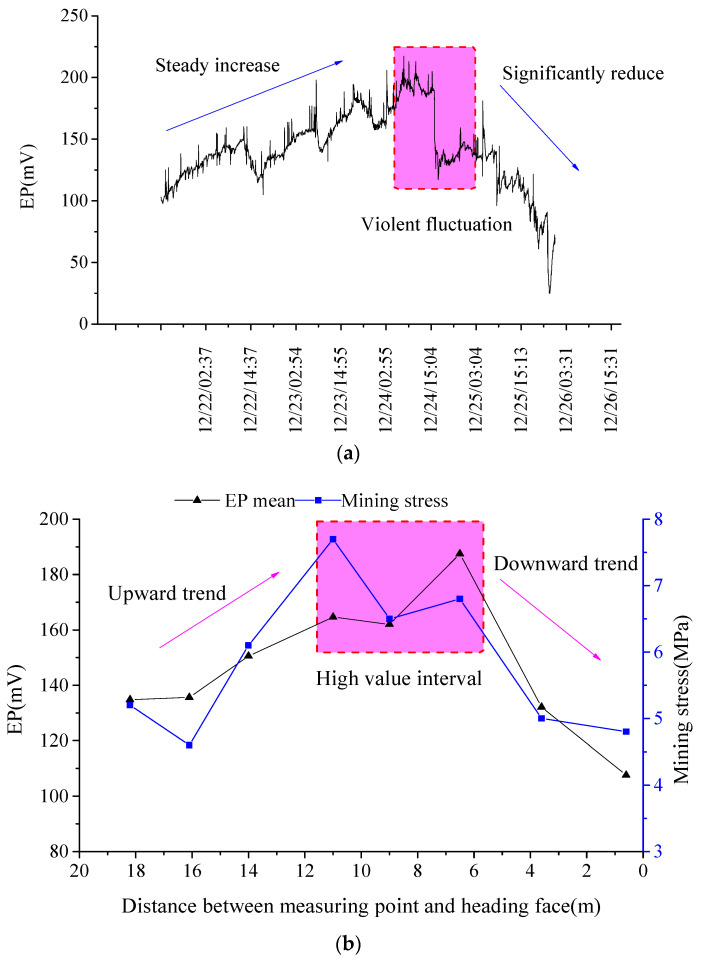
Measuring results of EP signal and mining stress ahead of the driving face, along with driving distance. (**a**) EP temporal response; (**b**) EP mean values and mining stress.

**Figure 4 ijerph-19-08949-f004:**
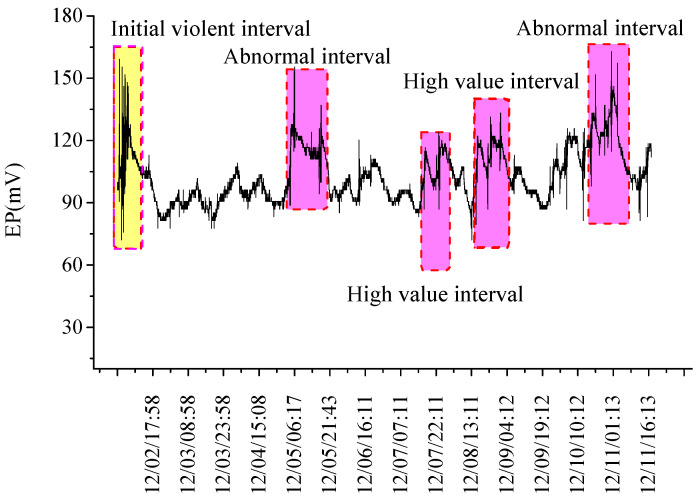
Measuring results of EP signal behind the driving face along with driving distance.

**Figure 5 ijerph-19-08949-f005:**
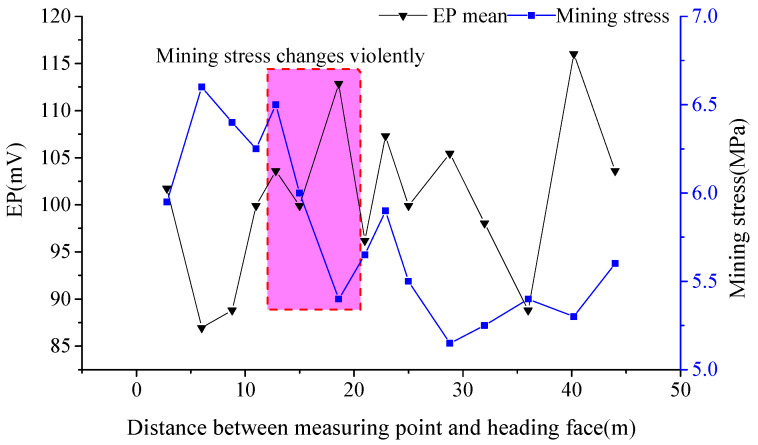
Measuring results of EP average value and mining stress behind the driving face, along with driving distance.

**Figure 6 ijerph-19-08949-f006:**
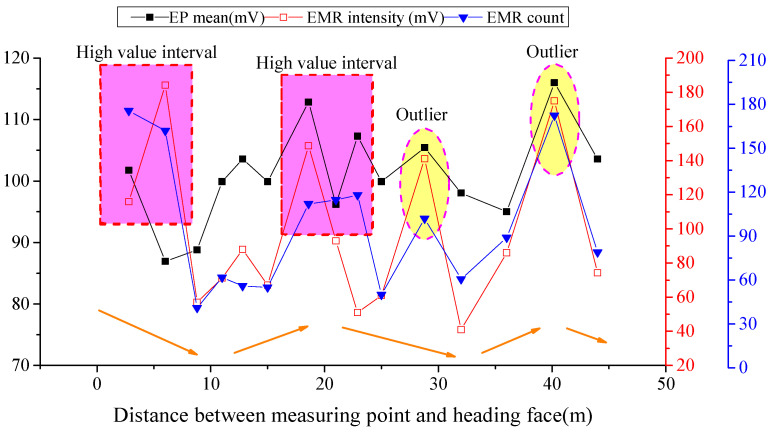
Measuring results of EP and EMR, along with driving distance behind the driving face.

**Figure 7 ijerph-19-08949-f007:**
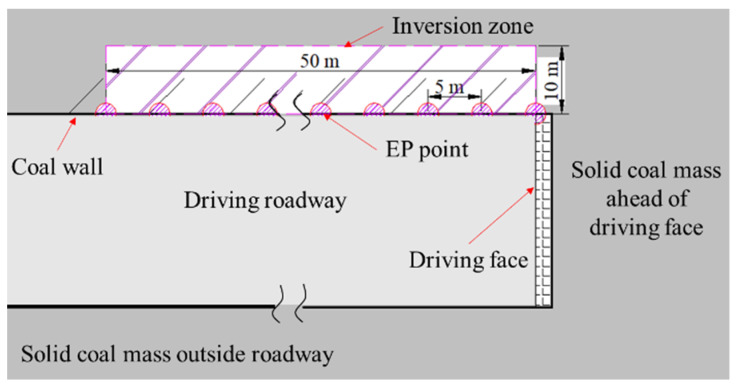
Schematic diagram of unilateral inversion EP measuring points arrangement of coal wall in the driving roadway.

**Figure 8 ijerph-19-08949-f008:**
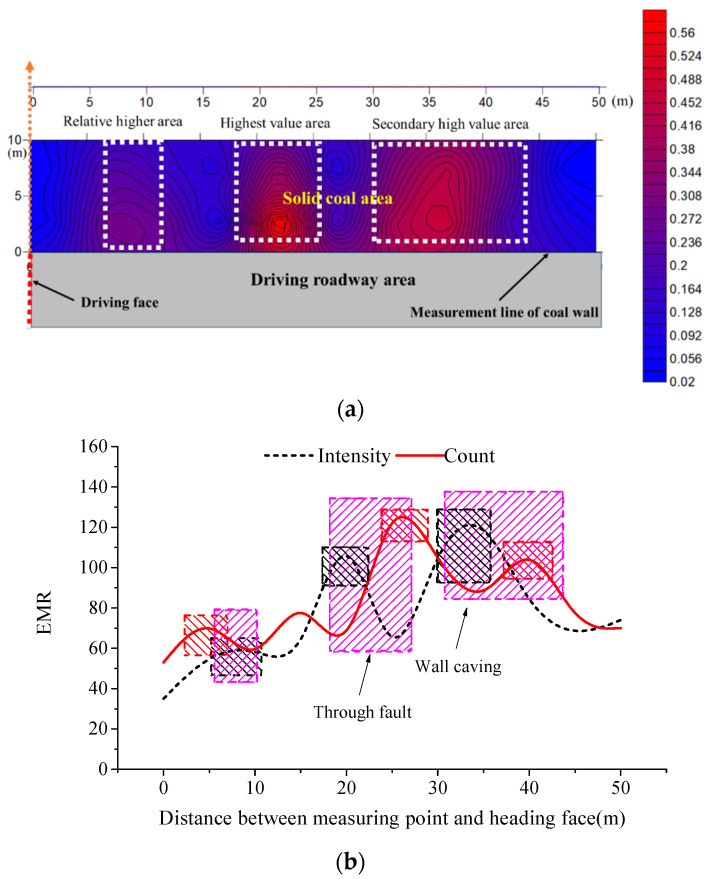
EP inversion results and EMR monitoring results behind the driving distance. (**a**) EP inversion distribution cloud map; (**b**) EMR indexes of intensity and count.

## Data Availability

The datasets analyzed or generated during the study can be obtained from the corresponding author.

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
