# Peer review of "Response Characteristics of Electric Potential and Its Relationship with Dynamic Disaster during Mining Activities: A Case Study in Xuehu Coal Mine, China"

_ijerph, 2022, doi:10.3390/ijerph19158949_

Round 1

Reviewer 1 Report

This article may be of great interest for the planning and execution of underground coal mining since it evaluates the relationship between electrical potential and underground coal mining activity. The authors present material of acceptable quality, however, some aspects should be improved in order to better understand the methodology, and better describe the results and their discussion.

Most of the figures should be improved, following the suggestions made. The methodology section should also be improved.

The document with annotations is attached.

Author Response

Thanks sincerely for your comments, which are very helpful for revising and improving our manuscript. To be honest, there are still some items and shortages in the manuscript. We have read your comments carefully which is valuable to improve the manuscript in-depth. Here we will make a further detailed reply to each comment and modify the manuscript substantially.

The methodology, results and discussion have been modified in the revised manuscript marked red. Besides, the figures have been improved with the guidance of related comments in the PDF.

Hope it will meet the requirements. If there are some unclear explain or missed items, please do not hesitate to contact us for further modification.

Reviewer 2 Report

The authors analyzed the response characteristics of electric potential signals during mining activities in a deep coal seam setting and also explored the relationship between electric potential and dynamic hazards such as gas and coal outburst. There are a few issues that need to be considered:

1. The Introduction section can be improved. I think discussing more related case studies in this section would be helpful.

2. The authors experimented with their methods using a deep coal seam setting. What about shallow coal mines? Would this still work? Did the authors conduct any sensitivity analysis?

3. The entire writing of the manuscript needs to be checked again. For example, the very first sentence in the Introduction section needs to be rewritten. Did the author mean "Coal is one of the major energy sources"? Issues like this were found often throughout the manuscript.

Author Response

Thanks sincerely for your comments, which are very helpful for revising and improving our manuscript. We have read the comments carefully which is valuable to improve the manuscript in-depth. Here we will make a further detailed reply to each comment and modify the manuscript substantially.
